# Differential distribution of eicosanoids and polyunsaturated fatty acids in the *Penaeus monodon* male reproductive tract and their effects on total sperm counts

Pisut Yotbuntueng[1], Surasak Jiemsup[1], Pacharawan Deenarn[1], Punsa Tobwor[1], Suganya Yongkiettrakul[1], Vanicha Vichai[1], Thapanee Pruksatrakul[1], Kanchana Sittikankaew[1], Nitsara Karoonuthaisiri[1,2,3], Rungnapa Leelatanawit[1¤], Wananit Wimuttisuk ORCID[1]*

1 National Center for Genetic Engineering and Biotechnology (BIOTEC), National Science and Technology Development Agency (NSTDA), Khlong Luang, Pathum Thani, Thailand, 2 Institute for Global Food Security, Queen's University, Belfast, United Kingdom, 3 International Joint Research Center on Food Security, Khlong Luang, Pathum Thani, Thailand

¤ Current address: Thermo Fisher Scientific (Thailand), Wattana, Bangkok, Thailand
* wananit.wim@biotec.or.th

## Abstract

Eicosanoids, which are oxygenated derivatives of polyunsaturated fatty acids (PUFAs), serve as signaling molecules that regulate spermatogenesis in mammals. However, their roles in crustacean sperm development remain unknown. In this study, the testis and vas deferens of the black tiger shrimp *Penaeus monodon* were analyzed using ultra-high performance liquid chromatography coupled with Orbitrap high resolution mass spectrometry. This led to the identification of three PUFAs and ten eicosanoids, including 15-deoxy-$\Delta^{12,14}$-prostaglandin $J_2$ (15d-PGJ$_2$) and (±)15-hydroxyeicosapentaenoic acid ((±)15-HEPE), both of which have not previously been reported in crustaceans. The comparison between wild-caught and domesticated shrimp revealed that wild-caught shrimp had higher sperm counts, higher levels of (±)8-HEPE in testes, and higher levels of prostaglandin $E_2$ (PGE$_2$) and prostaglandin $F_{2\alpha}$ in vas deferens than domesticated shrimp. In contrast, domesticated shrimp contained higher levels of (±)12-HEPE, (±)18-HEPE, and eicosapentaenoic acid (EPA) in testes and higher levels of 15d-PGJ$_2$, (±)12-HEPE, EPA, arachidonic acid (ARA), and docosahexaenoic acid (DHA) in vas deferens than wild-caught shrimp. To improve total sperm counts in domesticated shrimp, these broodstocks were fed with polychaetes, which contained higher levels of PUFAs than commercial feed pellets. Polychaete-fed shrimp produced higher total sperm counts and higher levels of PGE$_2$ in vas deferens than pellet-fed shrimp. In contrast, pellet-fed shrimp contained higher levels of (±)12-HEPE, (±)18-HEPE, and EPA in testes and higher levels of (±)12-HEPE in vas deferens than polychaete-fed shrimp. These data suggest a positive correlation between high levels of PGE$_2$ in vas deferens and high total sperm counts as well as a negative correlation between (±)12-HEPE in both shrimp testis and vas deferens and total sperm counts. Our analysis not only confirms the presence of PUFAs and eicosanoids in crustacean male reproductive organs, but also

**Data Availability Statement:** All relevant data are within the manuscript and its Supporting Information files.

**Funding:** This research has received funding supports from the National Center for Genetic Engineering and Biotechnology, Thailand [grant numbers P-17-50566 to WW]; the National Science and Technology Development Agency [grant number P14-50357 to RL], and the NSRF via the Program Management Unit for Human Resources & Institutional Development, Research and Innovation [grant number B05F640184 to WW]. The funders had no role in study design, data collection and analysis, decision to publish, or preparation of the manuscript.

**Competing interests:** The authors have declared that no competing interests exist.

suggests that the eicosanoid biosynthesis pathway may serve as a potential target to improve sperm production in shrimp.

## Introduction

Eicosanoids, which are derivatives of polyunsaturated fatty acids (PUFAs), serve as signaling molecules to regulate various physiological processes, including inflammation, immunity, and reproduction [1–3]. In mammals, eicosanoids have been shown to affect testicular development, sperm concentration, sperm motility, and infertility [4–6]. For instance, 15-deoxy-$\Delta^{12,14}$-prostaglandin $J_2$ (15d-PGJ$_2$) regulates the contraction of peritubular cells in the testis and may be involved in infertility in humans, while incubation of human spermatozoa in 1 μM prostaglandin $E_2$ (PGE$_2$) or 1 μM prostaglandin $F_{2\alpha}$ (PGF$_{2\alpha}$) improved sperm motility [4, 6].

The eicosanoid biosynthesis pathway in marine invertebrates utilizes eicosapentaenoic acid (EPA) and docosahexaenoic acid (DHA) as major substrates rather than arachidonic acid (ARA), which is predominantly used as eicosanoid precursors in mammals [7]. Nevertheless, ARA derivatives, namely PGE$_2$ and PGF$_{2\alpha}$, have been identified in the black tiger shrimp *Penaeus monodon*, the crab *Oziotelphusa senex senex*, the kuruma prawn *Marsupenaeus japonicus*, and the Florida crayfish *Procambarus paeninsulanus* [8–12]. In the crab *Carcinus maenas*, PGE$_2$, thromboxane $B_2$, and 6-keto-PGF$_{1\alpha}$ along with six ARA-derived hydroxy fatty acids, namely 5-, 8-, 9-, 11-, 12-, and 15-hydroxyeicosatetraenoic acids (HETEs), were detected in haemocytes [13]. Similarly, 12-HETE was identified in the hemolymph of *M. japonicus* [14]. Five oxygenated products of EPA, namely 5-, 8-, 9-, 12-, and 18-hydroxyeicosapenaenoic acids (HEPEs), were identified in the Pacific krill *Euphausia pacifica* [15]. Characterization of the eicosanoid biosynthesis pathway in crustaceans has thus far focused mostly on its roles in female reproductive maturation [8–12]. The eicosanoids involved in crustacean male reproduction have yet to be investigated in similar depth.

There has been limited information regarding the roles of eicosanoids in crustacean sperm development. A study in wild *Litopenaeus occidentalis* revealed that the administration of ibuprofen, which inhibits prostaglandin biosynthesis, increased normal spermatophore development [16]. This suggests a negative correlation between prostaglandin biosynthesis pathway and spermatogenesis in shrimp. On the other hand, high levels of dietary polyunsaturated fatty acids (PUFAs) showed a positive impact on crustacean sperm production [17, 18].

To further explore the roles of eicosanoids and PUFAs in crustacean spermatogenesis, *P. monodon* testes and vas deferens were subjected to liquid-liquid extraction and ultra-high performance liquid chromatography coupled with Orbitrap high resolution mass spectrometry (UHPLC-HRMS/MS) analysis. Levels of eicosanoids and PUFAs in testes and vas deferens were then compared between those of wild-caught and domesticated shrimp, which had high and low sperm counts, respectively. The effects of shrimp feed on eicosanoid and PUFA profiles in testes and vas deferens of domesticated shrimp were also examined. Our findings confirm the presence of eicosanoids in shrimp male reproductive tract and suggest that the roles of eicosanoids in regulating total sperm number in crustaceans are conserved relative to mammals.

## Materials and methods

### Ethical statement

All experiments were approved by the Institutional Animal Care and Use Committee of the National Center for Genetic Engineering and Biotechnology, Thailand (Approval Code

BT-Animal 13/2560). This permit covered the purchase wild-caught shrimp, shrimp transportation, shrimp rearing experiment, and shrimp dissection. No permit was required for the collection site access as the wild-caught broodstock collection from the Andaman Sea was conducted by local fishermen and purchased through a local shrimp farm. All experiments were performed in accordance with Animal Research: Reporting of *In Vivo* Experiments (ARRIVE) and conformed with international and national legal and ethical requirements, including the U.K. Animals (Scientific Procedures) Act, 1986 and associated guidelines, EU Directive 2010/63/EU for animal experiments, and the National Research Council's Guide for the Care and Use of Laboratory Animals.

## Shrimp sources

Wild-caught male shrimp were captured from the Andaman Sea, Thailand (salinity level at approximately 31 ppm) ($N = 10$). Eleven-month-old domesticated male *P. monodon*, which had been raised in earthen ponds and fed with commercial feed pellets, were acquired from the Shrimp Genetic Improvement Center (SGIC), Surat Thani, Thailand ($N = 10$). Average body weights of wild-caught and domesticated shrimp were $86.9 \pm 9.0$ and $66.8 \pm 7.6$ g, respectively. Shrimp testes and vas deferens were dissected and flash frozen in liquid $N_2$ for the quantification of eicosanoids and PUFAs using UHPLC-HRMS/MS. Spermatophores were collected and used for total sperm counts.

## Effects of shrimp feed

To determine changes in eicosanoid and PUFA levels in shrimp fed with different diets, eleven-month-old, domesticated males from the SGIC were fed with either polychaetes or feed pellets for four weeks ($N = 8$ each). Fatty acid profiles in polychaetes and feed pellets ($N = 4$ per feed) were analyzed using gas chromatography coupled with flame ionization detector (GC-FID) by the Nutrition Service at Central Lab Co., Ltd. (Thailand) (www.centrallabthai.com). Shrimp testes and vas deferens were dissected and flash frozen in liquid $N_2$. Spermatophores were collected and used to determine total sperm counts and percentage of sperm abnormality.

## Total sperm counts and sperm abnormality

Spermatophores were individually homogenized in a calcium-free sea water solution. After debris sedimentation, sperms were counted using a hemocytometer under a light microscope [19]. Abnormal sperms were defined as sperms with a misshaped head or tail as well as sperms with no head or tail [20]. Total sperm counts and abnormal sperm counts were determined from both spermatophores of each shrimp using average counts of four aliquots from each spermatophore homogenate. The percentage of abnormal sperm were then calculated based the percentage of abnormal sperm from the number of total live sperm.

## Chemicals and reagents

Eicosanoid standards were purchased from Cayman Chemicals (Michigan, USA). Standard compounds include prostaglandin $D_2$ ($PGD_2$), prostaglandin $E_1$ ($PGE_1$), $PGE_2$, $PGF_{2\alpha}$, 15d-$PGJ_2$, (±)5-hydroxy-6E,8Z,11Z,14Z-eicosatetraenoic acid ((±)5-HETE), (±)8-hydroxy-5Z,9E,11Z,14Z-eicosatetraenoic acid ((±)8-HETE), (±)9-hydroxy-5Z,7E,11Z,14Z-eicosatetraenoic acid ((±)9-HETE), (±)11-hydroxy-5Z,8Z,12E,14Z-eicosatetraenoic acid ((±)11-HETE), 12(R)-hydroxy-5Z,8Z,10E,14Z-eicosatetraenoic acid (12(R)-HETE), (±)5-hydroxy-6E,8Z,11Z,14Z,17Z-eicosapentaenoic acid ((±)5-HEPE), (±)8-hydroxy-5Z,9E,11Z,14Z,17Z-eicosapentaenoic acid ((±)8-HEPE), (±)9-hydroxy-5Z,7E,11Z,14Z,17Z-eicosapentaenoic acid

$((\pm)9\text{-HEPE})$, $(\pm)12$-hydroxy-5Z,8Z,10E,14Z,17Z-eicosapentaenoic acid $((\pm)12\text{-HEPE})$, $(\pm)$ 15-hydroxy-5Z,8Z,11Z,13E,17Z-eicosapentaenoic acid $((\pm)15\text{-HEPE})$, $(\pm)18$-hydroxy-5Z,8Z,11Z,14Z,16E-eicosapentaenoic acid $((\pm)18\text{-HEPE})$, ARA, DHA, and EPA. Deuterated compounds, namely $PGE_2\text{-}d_4$, 5(S)-HETE-$d_8$, 12(S)-HETE-$d_8$, and EPA-$d_5$, were used as internal standards to determine percent recovery during chemical extraction and during UHPLC-HRMS/MS analysis. All solvents and chemicals used in this study were HPLC grade or higher. Glacial acetic acid, acetonitrile, methanol, and ethanol were purchased from Merck (Darmstadt, Germany). Formic acid and cyclohexane were purchased from Fisher Scientific (Loughborough, UK). Hexane was purchased from J.T. Baker (New Jersey, USA). Ethyl acetate was purchased from Mallinckrodt Baker (New Jersey, USA). Isopropanol was purchased from RCI labscan (Bangkok, Thailand). Butylated hydroxytoluene (BHT) and Hank's Balanced Salt Solution (HBSS) were purchased from Sigma-Aldrich (Missouri, USA). Ethylenediaminetetra-acetic acid (EDTA) was purchased from Fluka (Steinheim, Switzerland). Water was purified by Barnstead GenPure Pro (Thermo Fisher Scientific, Massachusetts, USA).

## Sample preparation

Shrimp testes and vas deferens were individually homogenized in liquid $N_2$ and diluted in HBSS to adjust tissue concentration to 0.1 g/mL (wet weight). Organ homogenates were divided into 500 μL aliquots and adjusted to pH 4.0 using 5 μL of glacial acetic acid. Ten microliters of 10% BHT in HPLC-grade ethanol (w/v) were added as an antioxidant. Internal standards, including $PGE_2\text{-}d_4$, 5(S)-HETE-$d_8$, and EPA-$d_5$, were added to determine the percent recovery in each sample. An optimal extraction method was selected for each organ based on the recovery yields of the internal standards (S1 Table).

## Ethyl acetate extraction

Five hundred microliters of testis homogenates were subjected to ethyl acetate extraction at a 1:1 ratio (v/v) of tissue homogenate to ethyl acetate. Extraction mixtures were shaken in the dark for 15 min at 290 rpm and spun down at 8,000 rpm (8,228 ×g) for 10 min at 20˚C. The organic phase (upper phase) was collected, and the extraction process was repeated one more time. The extracts were evaporated to dryness and dissolved with 200 μL of 100% HPLC-grade ethanol for UHPLC-HRMS/MS analysis.

## Methanol-chloroform extraction

Five hundred microliters of vas deferens homogenates were subjected to methanol-chloroform extraction using the procedure modified from Folch extraction method [21]. Tissue homogenates were sequentially mixed with 3.75 mL of methanol, 6.25 mL of chloroform, and 3.12 mL of water. Samples were mixed rigorously for 1 min after each solvent was added. The mixture was shaken for 15 min at 290 rpm at room temperature and spun down at 8,000 rpm at 20˚C for 10 min. The organic phase (lower phase) was collected in a clean tube. The extraction was repeated by adding 3.75 mL of chloroform to the remaining aqueous phase. The mixture was vortexed for 1 min, shaken for 15 min at 290 rpm, and then spun down at 8,000 rpm at 20˚C for 10 min. The organic phase was collected, pooled, dried, and dissolved with 200 μL of 100% HPLC-grade ethanol for UHPLC-HRMS/MS analysis.

## UHPLC-HRMS/MS analysis

Chromatographic separation was performed on a Dionex UltiMate 3000RS UHPLC system (Thermo Fisher Scientific) with an Acclaim[TM] RSLC 120 C18 column (2.1×150 mm, 2.2 μm;

Thermo Fisher Scientific) under gradient conditions using mobile phase A (0.01% (v/v) acetic acid in water) and B (0.01% (v/v) acetic acid in acetonitrile) as previously described [22]. The linear gradient went from 30% B to 100% B within 17 min, followed by holding 100% B for 2 min. The elution gradient was returned to the starting condition of 30% B within 0.5 min and kept constant for 3.5 min before starting the next injection. UHPLC conditions included setting auto-sampler temperature at 10˚C, column temperature at 40˚C, injection volume at 5 μL, and flow rate at 300 μL/min for a total run time of 23 min.

Mass spectrometry analyses were performed on an Orbitrap Fusion™ Tribrid™ Mass Spectrometer (Thermo Scientific), equipped with electrospray ionization (ESI) source, and operated in negative ion mode. The mass spectrometer was controlled by the Xcalibur software (version 4.4.16.14) and calibrated using the ESI negative ion calibration solution (Pierce® LTQ velos ESI negative ion calibration) according to the manufacturer's protocol. Conditions for the mass spectrometer were set with the ESI voltage at 2,500 V in negative mode. Nitrogen was used as the sheath gas at 40 psi and as the auxiliary gas at 12 psi. Ultra-pure helium was used as the collision gas with the ion transfer tube temperature at 333˚C. The vaporizer temperature was 317˚C. Fragment ions of PUFA and eicosanoid standards were detected by the Orbitrap analyzer operated under target mass resolution of 120,000 with an automatic gain control (AGC) setting of $5 \times 10^4$ and a maximum ion injection time of 250 ms. The time-scheduled parallel reaction monitoring (PRM) method was used for data acquisition. Analytical characteristics of PUFA and eicosanoid standards used to identify and quantify the compounds in *P. monodon* tissues are provided in S2 Table. Both limit of detection (LOD) and limit of quantification (LOQ) were calculated based on the standard deviation (SD) of the response as well as the slope [23, 24].

$$\text{LOD} = 3.3 \times \frac{\text{SD}}{(\text{slope of the regression line})}$$

$$\text{LOQ} = 10 \times \frac{\text{SD}}{(\text{slope of the regression line})}$$

SD represents standard deviation of a blank sample with very low concentration (0.24–7.81 nM) of the measurand.

## Data processing and data analysis

Extracted-ion chromatograms (XIC) and mass spectra of eicosanoids and PUFAs obtained from the UHPLC-HRMS/MS analysis were processed and interpreted using Quan Browser (4.3.73.11), Xcalibur software (version 4.3.16.14). The area-under-the-curve (AUC) ratio of each metabolite was calculated by dividing the AUC of chromatographic peak of each respective metabolite with the AUC of the internal standard (12(S)-HETE-d$_8$). A pivot table of metabolite AUC ratios was constructed using Pandas (version 1.1.3, http://pandas.pydata.org), Python package [25, 26]. A heat map illustrating the AUC ratios of the metabolites was generated using Matplotlib (version 3.3.2, https://matplotlib.org/) and Seaborn (version 0.11.0, https://seaborn.pydata.org/), Python package [27, 28]. AUC ratios were converted to amounts of metabolites (ng/g tissue) using standard equations shown in S2 Table.

## Statistical analysis

Significant differences between the means of independent samples from the two sets of samples were assessed using the *t*-test with the threshold for significance set at $P < 0.05$ (*, † and #) or $P < 0.01$ (**, †† and ##).

## Results

### Comparison between wild-caught and domesticated males

Wild-caught male *P. monodon* broodstocks were captured from the Andaman Sea, Thailand (Fig 1A). Shrimp body weight and body length were recorded prior to dissection to obtain testes, vas deferens, and spermatophores (Fig 1B–1D). Similarly, domesticated males were obtained from SGIC, a biosecure facility located in Surat Thani Province, Thailand (Fig 1E). Their testes, vas deferens, and spermatophores were also collected (Fig 1F–1H). It should be noted that all shrimp spermatophores were intact without melanization. Data analysis revealed that wild-caught shrimp had larger body weight (Fig 1I), longer body length (Fig 1J), and higher spermatophore weight (Fig 1K) than those of domesticated shrimp. Additionally, the total sperm counts of wild-caught shrimp were also higher than those of domesticated shrimp (Fig 1L).

### Identification of eicosanoids and PUFAs in testes and vas deferens of wild-caught and domesticated *P. monodon*

To determine eicosanoid and PUFA profiles in *P. monodon* male reproductive organs, shrimp testes and vas deferens were subjected to ethyl acetate extraction and methanol-chloroform extraction, respectively. The organ extracts were then analyzed using UHPLC-HRMS/MS as depicted in Fig 2. The identity of each metabolite was verified based on retention time, precursor ion, proposed fragment ion, and m/z distribution (S3 Table). Testes and vas deferens of

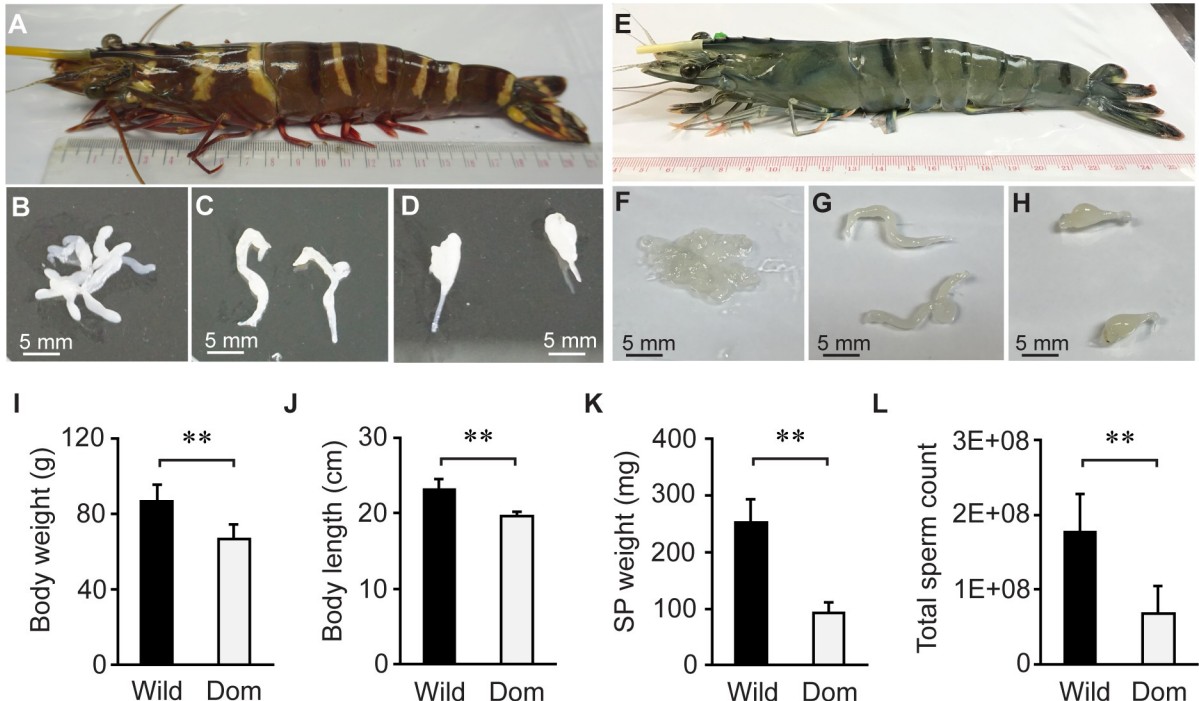

**Fig 1. Wild-caught shrimp had higher body weight, body length, spermatophore weight, and total sperm count than domesticated shrimp.** (A) Wild-caught shrimp ($N = 10$) were dissected to obtain (B) testes, (C) vas deferens, and (D) spermatophores. Dissection of (E) eleven-month-old, domesticated shrimp ($N = 10$) were also performed to collect (F) testes, (G) vas deferens, and (H) spermatophores for the analysis. Comparative analysis of (I) shrimp body weight, (J) body length, (K) spermatophore weight, and (L) total sperm count was performed between wild-caught (Wild; black bars) and domesticated shrimp (Dom; white bars). Error bars represent standard deviations. Asterisks indicate a significant difference between samples using the *t*-test (** for $P < 0.01$).

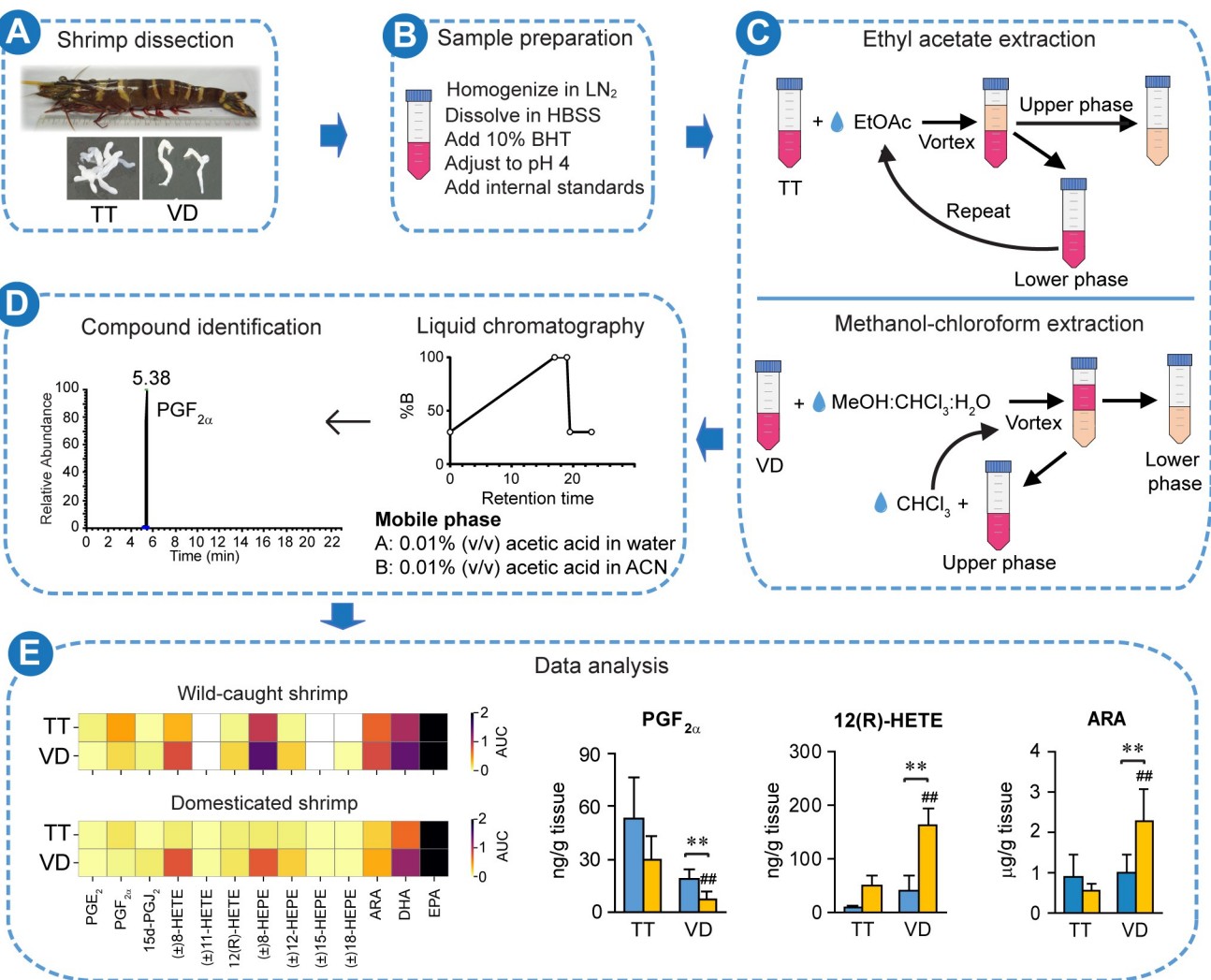

**Fig 2. Overview of liquid-liquid extraction and UHPLC-HRMS/MS analysis of eicosanoids and PUFAs in the *P. monodon* male reproductive tract.**
(A) Male *P. monodon* broodstocks were dissected to obtain testes (TT) and vas deferens (VD). (B) Sample preparation included tissue homogenization, pH adjustment, addition of antioxidant (10% BHT), and addition of internal standards. (C) Testis homogenates were subjected to ethyl acetate extraction (upper panel) whereas vas deferens homogenates were subjected to methanol-chloroform extraction (lower panel). (D) Tissue extracts were analyzed using the UHPLC system. Eicosanoids and PUFAs were then identified using HRMS/MS. (E) Metabolite quantification and data analysis were performed to determine levels of eicosanoids and PUFAs in each organ. $LN_2$ and ACN were abbreviated for liquid nitrogen and acetonitrile, respectively.

wild-caught and domesticated shrimp contained a combined number of 10 eicosanoids, including three prostaglandins ($PGE_2$, $PGF_{2\alpha}$, and 15d-$PGJ_2$), three HETEs ((±)8-, (±)11-, and 12(R)-HETEs), and four HEPEs ((±)8-, (±)12-, (±)15-, and (±)18-HEPEs) (Fig 3A–3J). Additionally, all three PUFAs, namely ARA, DHA, and EPA, were detected in all organ samples (Fig 3K–3M).

## Heat map visualization of eicosanoids and PUFAs in testes and vas deferens of wild-caught and domesticated shrimp

Heat map analysis was used to compare relative levels of eicosanoids and PUFAs based on the AUC ratio obtained from the UHPLC-HRMS/MS analysis (Fig 4). Testes of wild-caught shrimp contained seven eicosanoids, including $PGE_2$, $PGF_{2\alpha}$, 15d-$PGJ_2$, (±)8-HETE, 12(R)-

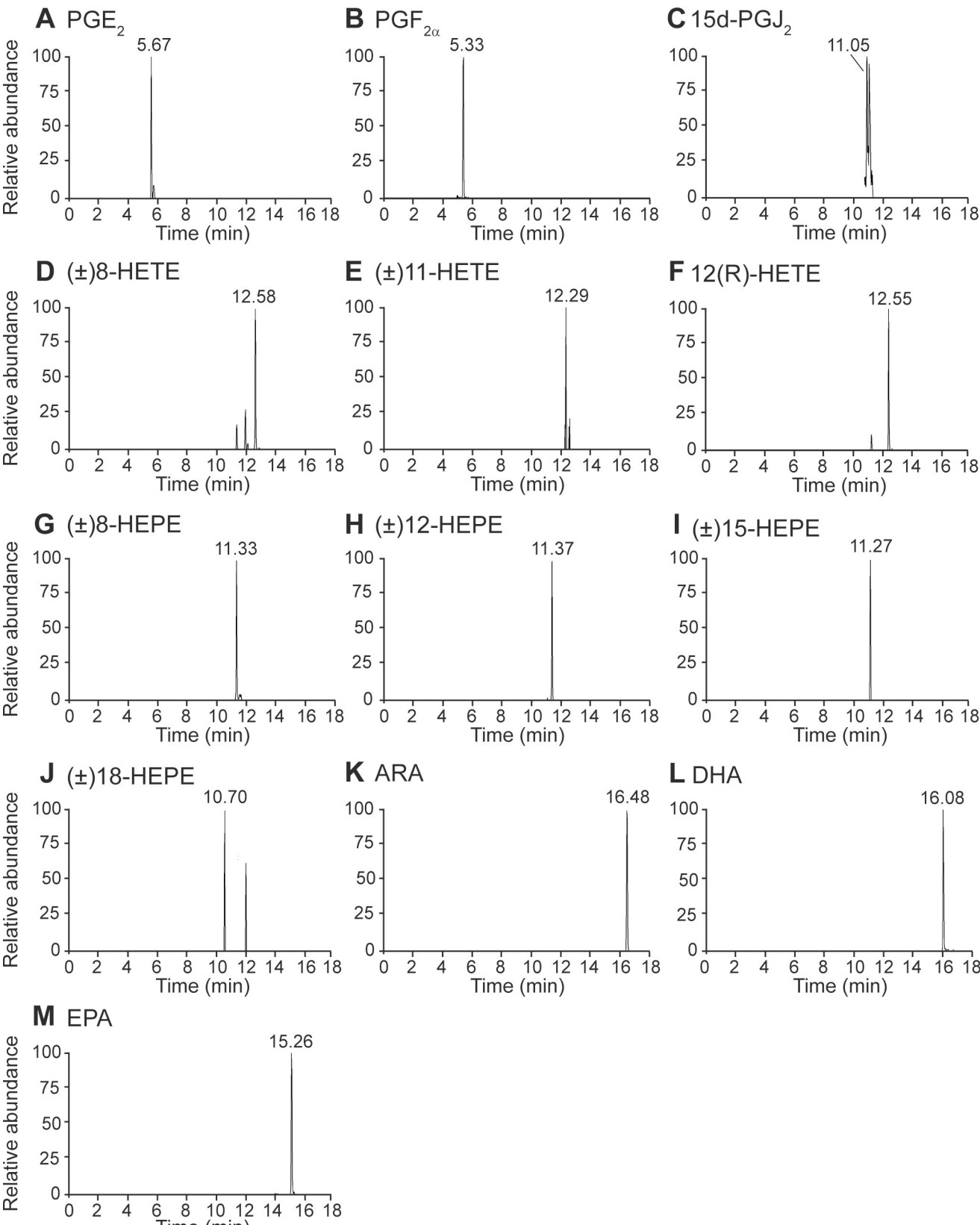

**Fig 3. Extracted-ion chromatogram (XIC) of eicosanoids and PUFAs identified in testes and vas deferens of wild-caught and domesticated *P. monodon*.** XIC of (A) PGE$_2$, (B) PGF$_{2\alpha}$, (C) 15d-PGJ$_2$, (D) (±)8-HETE, (E) (±)11-HETE, (F) 12(R)-HETE, (G) (±)8-HEPE, (H) (±)12-HEPE, (I) (±) 15-HEPE, (J) (±)18-HEPE, (K) ARA, (L) DHA, and (M) EPA were used to confirm the identities of the metabolites.

HETE, (±)8-HEPE, and (±)12-HEPE (Fig 4A, upper panel). Among these, PGF$_{2\alpha}$, (±)8-HETE, and (±)8-HEPE were present with high intensities in the heat map, suggesting that these eicosanoids may play crucial roles in spermatogenesis. Additionally, all three PUFAs were present in shrimp testes, in which ARA, DHA, and EPA were detected at low, medium, and high intensities relative to one another, respectively.

UHPLC-HRMS/MS analysis revealed that eight eicosanoids and three PUFAs were detected in vas deferens of wild-caught shrimp. In addition to the seven eicosanoids previously identified in testes, (±)18-HEPE was present in vas deferens with low intensities in the heat map (Fig 4A, lower panel). In contrast, (±)8-HETE and (±)8-HEPE were present with high intensities in vas deferens. Relative levels of ARA, EPA, and DHA in vas deferens were also similar to those in testes of wild-caught shrimp.

Heat map analysis of eicosanoids and PUFAs in testes and vas deferens of domesticated shrimp revealed different patterns from those in wild-caught shrimp. Three PUFAs and ten eicosanoids were detected in both testes and vas deferens of domesticated shrimp. The two additional eicosanoids identified only in domesticated shrimp were (±)11-HETE and (±) 15-HEPE, which were detected at low intensities in both testes and vas deferens. When relative levels of eicosanoids were examined in testes of domesticated shrimp, it was observed that all ten eicosanoids were present at relatively low intensities in the heat map, which was different from the pattern observed in testes of wild-caught shrimp. On the other hand, the heat map of vas deferens of domesticated shrimp displayed similar metabolic profiles to those of wild-caught shrimp, in which (±)8-HETE and (±)8-HEPE were major products of this pathway.

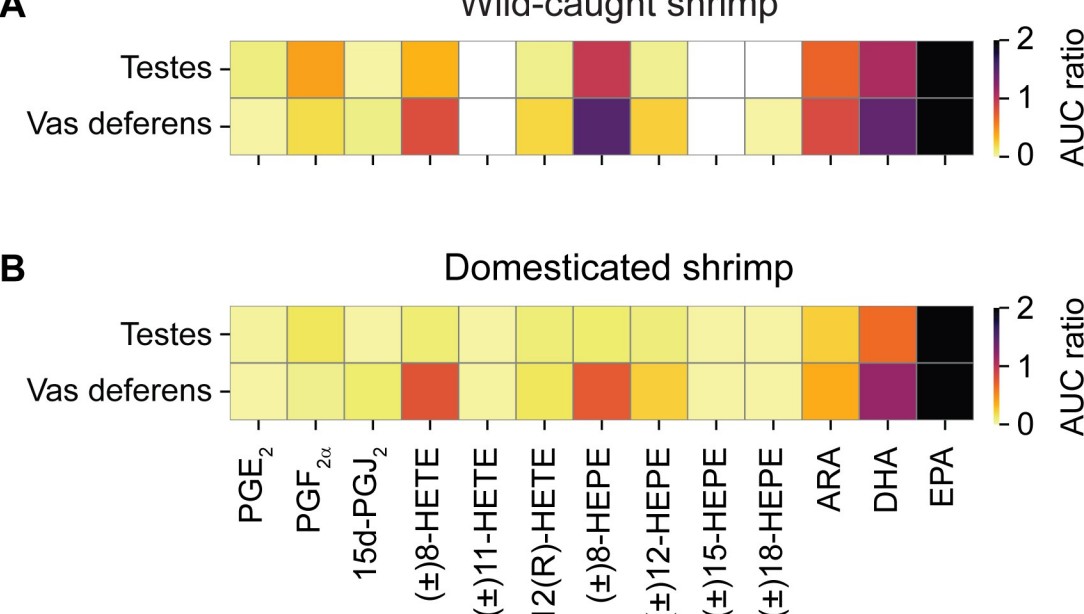

**Fig 4. Heat maps illustrating the presence and distribution of eicosanoids and PUFAs in testes and vas deferens of wild-caught and domesticated shrimp.** AUC ratio of each metabolite in (A) wild-caught and (B) domesticated shrimp was calculated using the AUC of the respective chromatographic peak divided by the AUC of the internal standard (12(S)-HETE-d$_8$). Metabolite intensities are displayed as colors ranging from yellow to black as shown in the color bar. White indicates that the metabolite was not detected.

Moreover, EPA was consistently the most abundant metabolite in testes and vas deferens of both wild-caught and domesticated shrimp, which illustrates the importance of EPA in the *P. monodon* sperm maturation process.

## Changes of eicosanoid and PUFA levels in the male reproductive tract

To follow metabolic changes that occurred during the sperm maturation process, levels of eicosanoids and PUFAs in shrimp testes were compared with those in vas deferens. In both wild-caught and domesticated shrimp, testes contained higher levels of $PGE_2$ (Fig 5A), but lower levels of 15d-$PGJ_2$ (Fig 5C), (±)8-HETE (Fig 5D), and (±)12-HEPE (Fig 5H) than vas deferens. On the other hand, levels of the remaining eicosanoids and PUFAs varied, depending on the shrimp source. In wild-caught shrimp, (±)18-HEPE (Fig 5I) was below the detection limit in testes but was detected at 4.18 ± 1.76 ng/g tissue in vas deferens. On the other hand, levels of $PGF_{2\alpha}$ (Fig 5B), (±)11-HETE (Fig 5E), 12(R)-HETE (Fig 5F), (±)8-HEPE (Fig 5G), ARA (Fig 5J), DHA (Fig 5K), and EPA (Fig 5L) were comparable between testes and vas deferens of wild-caught shrimp.

In domesticated shrimp, testes contained higher levels of $PGF_{2\alpha}$ (Fig 5B), but lower levels of 12(R)-HETE (Fig 5F), (±)8-HEPE (Fig 5G), ARA (Fig 5J), DHA (Fig 5K), and EPA (Fig 5L) than vas deferens. Interestingly, (±)11-HETE was detected only in vas deferens of domesticated shrimp (Fig 5E). As (±)11-HETE was below the limit of detection in vas deferens of wild-caught shrimp, it is likely that this metabolite is not essential for the sperm maturation process in *P. monodon*.

As wild-caught shrimp produced higher total sperm counts than domesticated shrimp, levels of eicosanoids and PUFAs in testes of wild-caught shrimp were compared to those in domesticated shrimp to determine correlations between these metabolites and total sperm counts. Testes of wild-caught shrimp contained higher levels of (±)8-HEPE (Fig 5G), but lower levels of (±)12-HEPE (Fig 5H), (±)18-HEPE (Fig 5I), and EPA (Fig 5L) than domesticated shrimp. On the other hand, levels of $PGE_2$ (Fig 5A), $PGF_{2\alpha}$ (Fig 5B), 15d-$PGJ_2$ (Fig 5C), (±)8-HETE (Fig 5D), 12(R)-HETE (Fig 5F), ARA (Fig 5J), and DHA (Fig 5K) in the testes of wild-caught and domesticated shrimp were comparable. Lastly, (±)11-HETE (Fig 5E) was not detected in testes in both wild-caught and domesticated shrimp, suggesting that this metabolite was not involved in shrimp spermatogenesis.

In vas deferens, wild-caught shrimp contained higher levels of $PGE_2$ (Fig 5A) and $PGF_{2\alpha}$ (Fig 5B), but lower levels of 15d-$PGJ_2$ (Fig 5C), (±)8-HETE (Fig 5D), (±)11-HETE (Fig 5E), 12(R)-HETE (Fig 5F), (±)12-HEPE (Fig 5H), ARA (Fig 5J), DHA (Fig 5K), and EPA (Fig 5L) than domesticated shrimp. Based on these data, it was deduced that high levels of (±)8-HEPE in testes and high levels of $PGE_2$ and $PGF_{2\alpha}$ in vas deferens are associated with high sperm counts. On the other hand, high levels of (±)12-HEPE, (±)18-HEPE, and EPA in testes and high levels of 15d-$PGJ_2$, (±)8-HETE, (±)11-HETE, 12(R)-HETE, (±)12-HEPE, and PUFAs in vas deferens are correlated with low sperm counts in *P. monodon*. Although (±)15-HEPE was identified in both the testes and vas deferens of domesticated shrimp as shown in the XIC (Fig 3I) and the heat map (Fig 4B), this metabolite was detected at the level above the limit of detection in only 2 out of 10 shrimp samples (S4 File). Therefore, (±)15-HEPE was excluded from the quantitative analysis.

## Effects of shrimp feed on eicosanoids and PUFAs in the male reproductive tract

In hatcheries, domesticated males are typically fed with live *Perinereis nuntia* polychaetes instead of commercial feed pellets to increase total sperm counts. To test the effects of shrimp

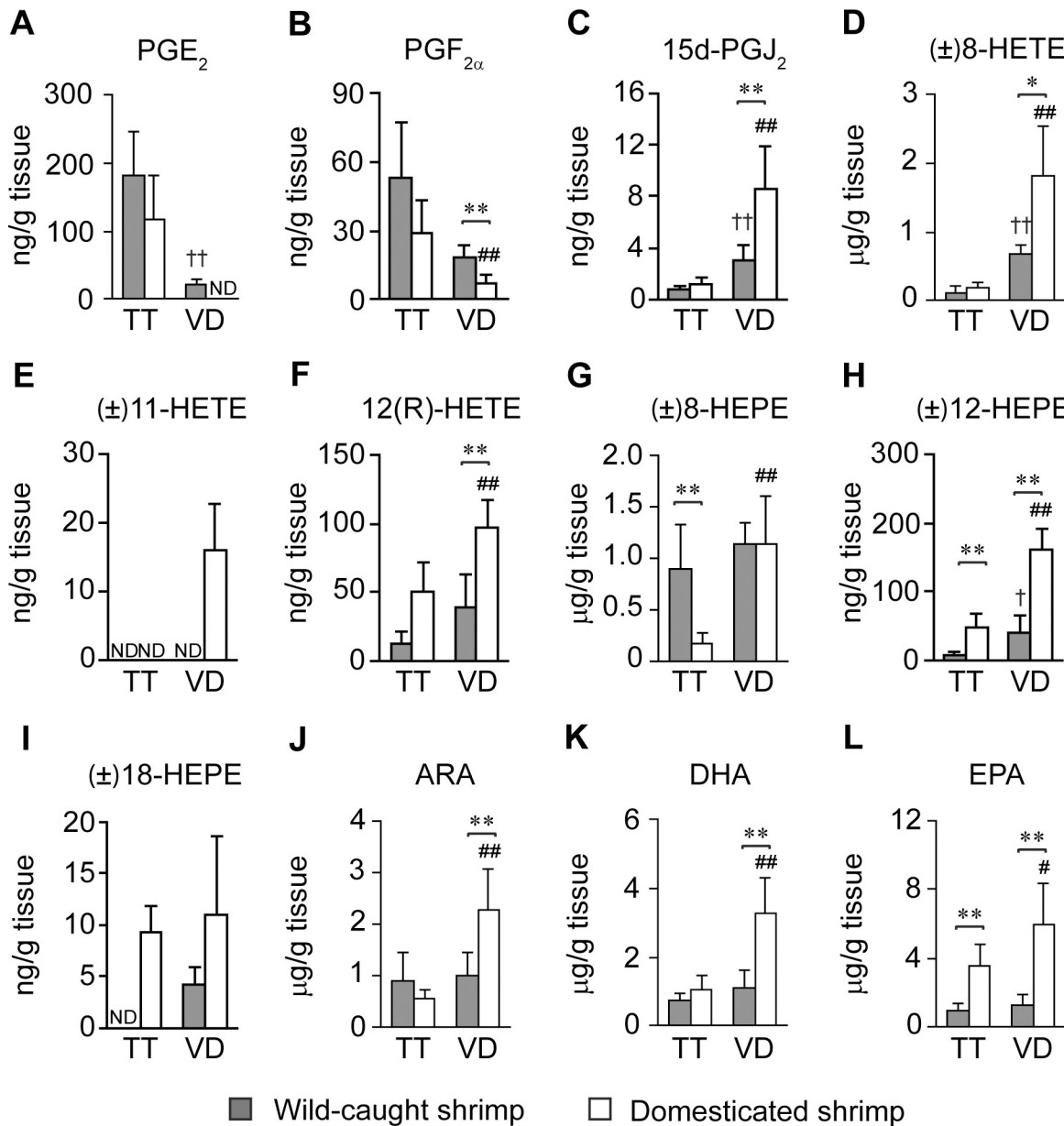

**Fig 5. Quantitative analysis of eicosanoids and PUFAs in testes and vas deferens of wild-caught and domesticated *P. monodon*.**
Levels of (A) PGE$_2$, (B) PGF$_{2\alpha}$, (C) 15d-PGJ$_2$, (D) (±)8-HETE, (E) (±)11-HETE, (F) 12(R)-HETE, (G) (±)8-HEPE, (H) (±)12-HEPE, (I) (±)18-HEPE, (J) ARA, (K) DHA, and (L) EPA in testes (TT) and vas deferens (VD) were compared between wild-caught (gray bar, $N = 6$) and domesticated shrimp (white bar, $N = 10$). Data are shown as means ± SD. Asterisks indicate statistically significant differences in metabolic levels between wild-caught and domesticated shrimp using the *t*-test (* for $P < 0.05$ and ** for $P < 0.01$). Daggers indicate statistically significant differences in metabolic levels between testes and vas deferens of wild-caught shrimp using the *t*-test († for $P < 0.05$ and †† for $P < 0.01$). Hashes indicate statistically significant differences in metabolic levels between testes and vas deferens of domesticated shrimp using the *t*-test (# for $P < 0.05$ and ## for $P < 0.01$). ND indicates that the designated metabolite was not detected in this analysis.

feed on PUFA and eicosanoid profiles in male reproductive tract, eleven-month-old, domesticated males from the same genetic background were fed with either polychaetes or feed pellets for four weeks. Polychaetes and feed pellets were analyzed using GC-FID, revealing that polychaetes contained higher levels of total saturated fatty acids, monounsaturated fatty acids, and

**Table 1. Fatty acid compositions in mg per g dry weight of polychaetes and feed pellets.**

| Common Name | Abbrev. | Fatty acid composition (mg/g dry weight) | |
|---|---|---|---|
| | | Polychaetes | Pellets |
| Myristic acid | C14:0 | 1.03 ± 0.18 | 1.43 ± 0.21 |
| Pentadecanoic acid | C15:0 | 0.49 ± 0.02 | 0.28 ± 0.04** |
| cis-10-Pentadecenoic acid | C15:1 | 2.10 ± 0.22 | ND |
| Palmitic acid | C16:0 | 34.48 ± 0.63 | 15.22 ± 1.95** |
| Palmitoleic acid | C16:1 | 3.57 ± 0.29 | 2.17 ± 0.21** |
| Heptadecanoic acid | C17:0 | 2.08 ± 0.10 | 0.55 ± 0.07** |
| cis-10-Heptadecenoic acid | C17:1 | 0.33 ± 0.03 | 0.11 ± 0.10* |
| Stearic acid | C18:0 | 11.21 ± 0.48 | 4.02 ± 0.51** |
| Elaidic acid | C18:1n9t | 4.68 ± 0.28 | ND |
| Oleic acid | C18:1n9c | 15.08 ± 0.12 | 14.66 ± 1.46 |
| Linoleic acid | C18:2n6c | 17.05 ± 1.32 | 10.04 ± 0.47** |
| Linolenic acid | C18:3n3 | 1.42 ± 0.14 | 0.59 ± 0.09** |
| Arachidic acid | C20:0 | 0.26 ± 0.22 | 0.47 ± 0.08 |
| cis-11-Eicosenoic acid | C20:1n9 | 3.16 ± 0.33 | 0.67 ± 0.08** |
| cis-11,14-Eicosadienoic acid | C20:2n6 | 7.17 ± 0.40 | ND |
| cis-8,11,14-Eicosatrienoic acid | C20:3n6 | 1.05 ± 0.03 | ND |
| Arachidonic acid (ARA) | C20:4n6 | 6.27 ± 0.32 | 0.05 ± 0.08** |
| cis-5,8,11,14.17-Eicosapentaenoic acid (EPA) | C20:5n3 | 6.33 ± 0.46 | 0.23 ± 0.20** |
| Heneicosanoic acid | C21:0 | 0.82 ± 0.05 | ND |
| Behenic acid | C22:0 | ND | 0.18 ± 0.31 |
| cis-4,7,10,13,16,19-Docosahexaenoic acid (DHA) | C22:6n3 | 2.66 ± 0.38 | 0.12 ± 0.20** |
| Tricosanoic acid | C23:0 | 1.03 ± 0.14 | ND |
| Lignoceric acid | C24:0 | ND | 0.08 ± 0.14 |
| n-3 highly unsaturated fatty acids (HUFA) | | 2.91 ± 0.16 | 0.30 ± 0.51** |
| Total saturated fatty acid (SFA) | | 51.79 ± 0.77 | 22.23 ± 2.42** |
| Total monounsaturated fatty acid (MUFA) | | 28.92 ± 0.42 | 17.61 ± 1.73** |
| Total polyunsaturated fatty acid (PUFA) | | 42.00 ± 1.72 | 11.02 ± 0.93** |
| Total trans fatty acid | | 4.68 ± 0.28 | ND |
| Total fatty acid | | 122.71 ± 0.96 | 50.86 ± 3.34** |

Fatty acid profiles in polychaetes and feed pellets were analyzed using GC-FID. ND is abbreviated for not detected. Asterisks indicate significant differences between the average values of fatty acids found in polychaetes and feed pellets with the threshold for significance set at $P < 0.05$ (*) or $P < 0.01$ (**).

polyunsaturated fatty acids, including ARA, EPA, and DHA, than feed pellets (Table 1). Shrimp fed with polychaetes also had higher sperm counts (Fig 6A), but comparable percentage of sperm abnormality to those of pellet-fed shrimp (Fig 6B).

## Quantitative analysis of eicosanoids and PUFAs in the testes and vas deferens of polychaete- and pellet-fed shrimp

To determine whether eicosanoid and PUFA profiles in the male reproductive tract were affected by shrimp diet, testes and vas deferens of polychaete- and pellet-fed shrimp were analyzed using UHPLC-HRMS/MS. First, levels of eicosanoids and PUFAs were compared between testes and vas deferens of shrimp in each feed group to determine metabolic changes during spermatogenesis and sperm maturation process, respectively. Data analysis revealed that the majority of the metabolites, including 15d-PGJ$_2$ (Fig 7C), (±)8-HETE (Fig 7D), 12(R)-

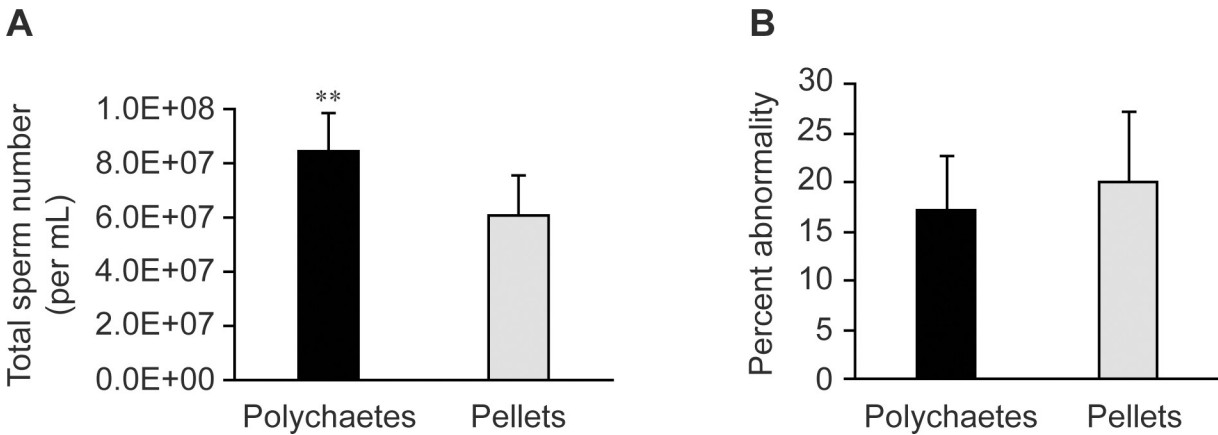

**Fig 6. Total sperm counts and percentage of sperm abnormality in domesticated shrimp fed with polychaetes and feed pellets.**
Spermatophores of domesticated shrimp fed with either polychaetes (black bar, $N = 8$) or commercial feed pellets (gray bar, $N = 8$) were used in the analysis to determine (A) total sperm counts and (B) percentage of sperm abnormality. Error bars represent standard deviations. Asterisks indicate a significant difference between samples using the $t$-test ($**$ for $P < 0.01$).

HETE (Fig 7F), ($\pm$)12-HEPE (Fig 7G), ($\pm$)18-HEPE (Fig 7H), ARA (Fig 7I), DHA (Fig 7J), and EPA (Fig 7K), were detected at higher levels in vas deferens than testes of shrimp in both feed groups, suggesting that these metabolites were more essential in the sperm maturation process than spermatogenesis. Meanwhile, levels of $PGF_{2\alpha}$ were comparable between testes and vas deferens of shrimp in both feed groups (Fig 7B). The two eicosanoids with distinct metabolic patterns according to feed types were $PGE_2$ and ($\pm$)11-HETE (Fig 7A and 7E). More specifically, levels of $PGE_2$ were comparable between testes and vas deferens of polychaete-fed shrimp (Fig 7A). In pellet-fed shrimp, however, $PGE_2$ was detected at similar levels in testes but became undetectable in vas deferens, suggesting that the use of feed pellets reduced the levels of $PGE_2$ in this organ. In contrast, ($\pm$)11-HETE was absent in most tested samples except in vas deferens of polychaete-fed shrimp (Fig 7E). These data suggest that although changes in shrimp diet did not alter relative levels of most PUFAs and eicosanoids in shrimp testes and vas deferens, the distribution of certain ARA-derived eicosanoids, namely $PGE_2$ and ($\pm$)11-HETE, in vas deferens was affected by shrimp feed. Lastly, ($\pm$)8-HEPE and ($\pm$)15-HEPE were excluded from the analysis as they were quantifiable in less than 50% of samples.

Results from other studies as well as data from our own analysis (Fig 6) revealed that polychaete-fed shrimp had higher total sperm counts than pellet-fed shrimp [18, 20] However, the effects of shrimp diets on levels of PUFAs and eicosanoids in shrimp testes and vas deferens have yet to be investigated. In this study, levels of eicosanoids and PUFAs in testes were compared between polychaete- and pellet-fed shrimp to assess the impact of shrimp feed. Testes of polychaete-fed shrimp contained higher levels of ($\pm$)8-HETE (Fig 7D), but lower levels of 15d-$PGJ_2$ (Fig 7C), ($\pm$)12-HEPE (Fig 7G), ($\pm$)18-HEPE (Fig 7H), ARA (Fig 7I), DHA (Fig 7J), and EPA (Fig 7K) than those in pellet-fed shrimp. On the other hand, levels of $PGE_2$ (Fig 7A), $PGF_{2\alpha}$ (Fig 7B), and 12(R)-HETE (Fig 7F) were comparable in testes of polychaete- and pellet-fed shrimp. Lastly, ($\pm$)11-HETE (Fig 7E) was not detected in testes of shrimp from both feed groups, indicating that this compound was not involved in shrimp spermatogenesis.

A similar analysis was performed to compare levels of eicosanoids and PUFAs in vas deferens between polychaete- and pellet-fed shrimp. The UHPLC-HRMS/MS analysis revealed that $PGE_2$ (Fig 7A) and ($\pm$)11-HETE (Fig 7E) were present only in vas deferens of polychaete-fed shrimp. As levels of these metabolites were below the limit of detection in vas deferens of pellet-fed shrimp, it is possible that the lack of these eicosanoids might be correlated with low

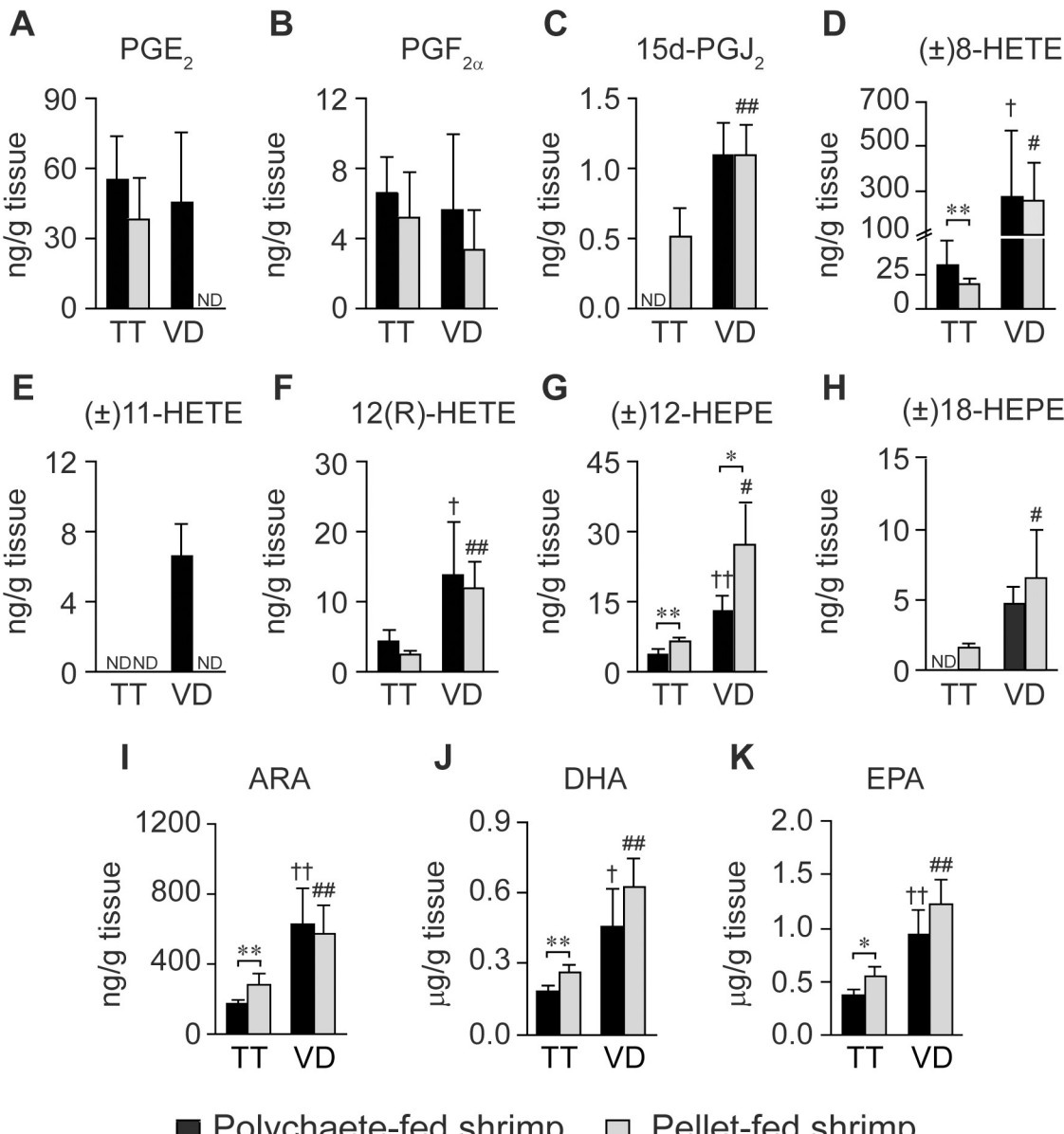

**Fig 7. Comparative analysis of eicosanoid and PUFA levels in testes and vas deferens of eleven-month-old, domesticated shrimp fed with polychaetes or feed pellets.** Levels of (A) $PGE_2$, (B) $PGF_{2\alpha}$, (C) 15d-$PGJ_2$, (D) (±)8-HETE, (E) (±)11-HETE, (F) 12(R)-HETE, (G) (±)12-HEPE, (H) (±)18-HEPE, (I) ARA, (J) DHA, and (K) EPA were compared between polychaete- (black bar, $N = 5$) and pellet-fed shrimp (gray bar, $N = 5$). Error bars represent standard deviations. Asterisks indicate statistically significant differences in metabolic levels between polychaete- and pellet-fed shrimp using the $t$-test (* for $P < 0.05$ and ** for $P < 0.01$). Daggers indicate statistically significant differences in metabolic levels between testes and vas deferens of polychaete-fed shrimp using the $t$-test († for $P < 0.05$ and †† for $P < 0.01$). Hashes indicate statistically significant differences in metabolic levels between testes and vas deferens of pellet-fed shrimp using the $t$-test (# for $P < 0.05$ and ## for $P < 0.01$). ND means that the designated metabolite was not detected.

sperm counts. On the other hand, levels of (±)12-HEPE (Fig 7G) were higher in vas deferens of pellet-fed shrimp than in those of polychaete-fed shrimp, suggesting a negative correlation between high levels of (±)12-HEPE and total sperm counts. Meanwhile, levels of $PGF_{2\alpha}$ (Fig 7B), 15d-$PGJ_2$ (Fig 7C), (±)8-HETE (Fig 7D), 12(R)-HETE (Fig 7F), (±)18-HEPE (Fig 7H), ARA (Fig 7I), DHA (Fig 7J), and EPA (Fig 7K) were comparable in vas deferens of polychaete-

and pellet-fed shrimp, indicating that the difference in shrimp feed did not affect the production of these eicosanoids in vas deferens.

## Discussion

Poor reproductive performance in domesticated males is one of the contributing factors that delay the progress of shrimp aquaculture industry [29, 30]. Although tremendous research efforts have been made to improve shrimp breeding, total sperm counts in domesticated males remain lower than those in wild-caught males [31, 32]. In fact, studies have shown that the reproductive success of penaeid shrimp depends on various factors, including shrimp age, shrimp size, genetic background, rearing environment, hormones, and nutrients [20, 33–36]. As dietary PUFAs have been shown to improve sperm quality in crustaceans [17, 37], it is likely that increasing PUFA consumption would affect levels of PUFAs and their downstream metabolites in the crustacean male reproductive tract. In this study, *P. monodon* testes and vas deferens were subjected to ethyl acetate and methanol-chloroform extraction, respectively. The organ extracts were then analyzed using UHPLC-HRMS/MS, revealing that a total of ten eicosanoids and three PUFAs were detected in shrimp testes and vas deferens. Correlations between metabolic profiles, organ types, and total sperm counts were then examined to assess the roles of PUFAs and eicosanoids in crustacean male reproduction.

### Spermatophore quality between wild-caught and domesticated crustaceans

Spermatophore quality of decapod crustaceans can be evaluated using several parameters, including melanization, spermatophore weight, sperm number, sperm viability, sperm abnormality, and spermatophore absence rates [35, 38]. The loss of spermatophore quality can be attributed to stress, poor nutrient, and the length of time spent in captivity for wild-caught shrimp [19, 39]. In this study, all spermatophores were present and no melanization was observed in all collected samples. Wild-caught shrimp had higher spermatophore weights and higher total sperm counts than domesticated shrimp, suggesting that the spermatophore quality of wild-caught shrimp was higher than those of domesticated shrimp in this study. Our data were supported by Rodríguez et al. (2007), in which the wild-caught Pacific white shrimp *Litopenaeus vannamei* produced higher total sperm counts than the domesticated counterparts [32]. However, other studies reported that spermatophore weights and total sperm counts of wild-caught and domesticated shrimp were comparable [18, 31]. The discrepancy between these studies may stem from the difference in shrimp size. A positive correlation between shrimp size and total sperm count has previously been reported in a different study in *L. vannamei* [40]. Upon closer examination of our data and the data from Rodríguez et al. (2007), it was confirmed that wild-caught males with higher body weights also had higher total sperm counts than domesticated males in both studies [32], whereas wild-caught and domesticated males with similar body weights also contained comparable total sperm counts [18, 31]. As a result, shrimp body weight should also be taken into consideration during the comparison of total sperm counts between shrimp from different sources.

### Correlations between levels of PUFAs in shrimp diets, shrimp male reproductive organs, and spermatogenesis

One of the contributing factors that affect sperm quality is the amounts of PUFAs in crustacean diets [41]. Supplementation of fish oil enriched in n-3 and n-6 PUFAs has been shown to increase levels of ARA, EPA, and DHA in *P. monodon* testis and enhance the number of spermatozoa in male broodstocks [41]. In fact, spermatophore quality can be used to determine the efficiency of crustacean maturation diets [18, 42, 43]. In this study, the consumption of

polychaetes, which contained higher levels of n-3 and n-6 PUFAs than feed pellets, did not result in higher levels of ARA, EPA, and DHA in shrimp testis and vas deferens than those of pellet-fed shrimp. Moreover, a negative correlation between levels of dietary PUFAs and levels of PUFAs in testis and vas deferens was observed, suggesting that aside from the dietary intake, other factors also influenced levels of PUFAs in crustacean male reproductive organs.

In the oriental river prawn *Macrobrachium nipponense*, a positive correlation between high levels of n-6 PUFAs in the testis and crustacean spermatogenesis has been reported [44]. Levels of EPA and DHA in the testis increased as shrimp progressed from early to mid and late stages of gonad development [44]. Nevertheless, this observation might be species-specific as there was no correlation between levels of EPA and DHA in the testis and spermatogenesis or mating activities in *M. rosenbergii* [45]. On the other hand, high levels of ARA have typically been correlated with low sperm counts and poor sperm motility in mammals [46]. However, the effects of high levels of ARA in male reproductive organ have never been reported in crustaceans. In this study, the analysis of wild-caught and domesticated shrimp revealed a negative correlation between total sperm counts and high levels of EPA in testes as well as high levels of ARA, EPA, and DHA in vas deferens. These data were supported by the analysis of polychaete- and pellet-fed shrimp, in which higher levels of EPA were observed in testes of pellet-fed shrimp than those of polychaete-fed shrimp.

## The identification of eicosanoids in the *P. monodon* male reproductive tract

As PUFAs are known precursors of eicosanoids, the increased levels of PUFAs in shrimp testis and vas deferens could potentially result in higher production of eicosanoids in these organs. In this study, the UHPLC-HRMS/MS analysis revealed that ten eicosanoids and three PUFAs were found in *P. monodon* testes and vas deferens. These included $PGE_2$, $PGF_{2\alpha}$, ($\pm$)8-HETE, ($\pm$)11-HETE, 12(R)-HETE, ($\pm$)8-HEPE, ($\pm$)12-HEPE, and ($\pm$)18-HEPE, all of which had previously been identified in crustaceans [8, 9, 11–13, 15, 47–49]. Additionally, to the best of our knowledge, this is also the first identification of 15d-$PGJ_2$ and ($\pm$)15-HEPE in crustaceans. The roles of 15d-$PGJ_2$ in male reproductive maturation has been firmly established in mammals [4, 50]. High levels of 15d-$PGJ_2$ in the testis and vas deferens were associated with impaired spermatogenesis in pigs and male infertility in humans, respectively [4, 50]. In the testis, 15d-$PGJ_2$ acted through the reactive oxygen species (ROS) pathway, which prevented the differentiation of human testicular peritubular cells [4]. This resulted in the loss of contractility of the peritubular cells of the testis, which led to impaired spermatogenesis. On the other hand, high levels of 15d-$PGJ_2$ in vas deferens activated the PPARγ pathway, which regulated luminal electrolytes in the reproductive ducts that affected sperm functions and viability [50]. As high levels of 15d-$PGJ_2$ were detected in vas deferens of *P. monodon*, we propose that excess levels of 15d-$PGJ_2$ might impair sperm function and viability in shrimp vas deferens, which subsequently result in low sperm counts in penaeid shrimp.

Although the roles of 15d-$PGJ_2$ in spermatogenesis are well-established in mammals, the function of 15-HEPE in testis and vas deferens has not been reported in any organism. Nevertheless, the inhibition of 15-lipoxygenase, which converts EPA to 15-HEPE, can improve sperm motility and acrosome reaction rates as well as reduce the oxidative stress via ROS pathway [51]. Therefore, the identification of 15-HEPE in testis and vas deferens of domesticated shrimp might also indicate that the ROS pathway may be activated in domesticated shrimp.

## Effects of eicosanoids in crustacean total sperm counts

In this study, the heat map analysis of relative abundance of PUFAs and eicosanoids in shrimp reproductive tract revealed that ($\pm$)8-HEPE and ($\pm$)8-HETE were the two most abundant eicosanoids in shrimp testes and vas deferens. In fact, high levels of ($\pm$)8-HETE and ($\pm$)

**A** **Metabolic changes as sperm travelled from testis to vas deferens**

**Wild-caught and domesticated shrimp**

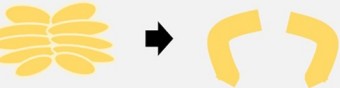

Testis    Vas deferens

**Polychaete- and pellet-fed shrimp**

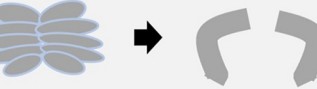

Testis    Vas deferens

Level increased

15d-PGJ$_2$, (±)8-HETE (±)12-HEPE

15d-PGJ$_2$, (±)8-HETE, 12(R)-HETE, (±)12-HEPE, (±)18-HEPE, ARA, DHA, EPA

Level decreased

PGE$_2$

None

**B** **Comparison between shrimp sources**

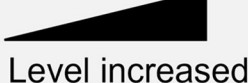

**High in wild-caught shrimp**

**Testis**    (±)8-HEPE

**Vas deferens**    PGE$_2$, PGF$_{2α}$

**High in domesticated shrimp**

**Testis**    (±)12-HEPE, (±)18-HEPE, EPA

**Vas deferens**    15d-PGJ$_2$, (±)8-HETE, (±)11-HETE, 12(R)-HETE, (±)12-HEPE, ARA, EPA, DHA

**C** **Comparison between shrimp feeds**

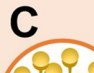

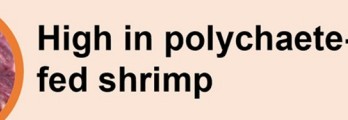

**High in polychaete-fed shrimp**

**Testis**    (±)8-HETE

**Vas deferens**    PGE$_2$, (±)11-HETE

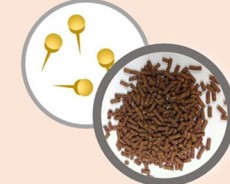

**High in pellet-fed shrimp**

**Testis**    15d-PGJ$_2$, (±)12-HEPE, (±)18-HEPE, ARA, DHA, EPA

**Vas deferens**    (±)12-HEPE

**Fig 8. Summary of changes in eicosanoid and PUFA profiles in testes and vas deferens of *P. monodon*.** (A) Metabolic changes that occurred as sperm travels from testes to vas deferens in wild-caught and domesticated shrimp as well as in shrimp fed with different diets. (B) Metabolic changes in testes and vas deferens of wild-caught and domesticated shrimp, which represent shrimp with high and low total sperm counts, respectively. (C) Metabolic changes in testes and vas deferens of shrimp fed with polychaetes and feed pellets, which also resulted in high and low total sperm counts, respectively. Metabolites that share the same correlation in both sets of samples (shrimp source and shrimp feed) are underlined.

8-HEPE were reported in *E. pacifica* [52] and high levels of (±)8-HEPE were also detected in the hepatopancreas of *P. monodon* [49], suggesting that these hydroxy fatty acids were major metabolites and ubiquitously expressed in crustaceans.

To assess the roles of eicosanoids in shrimp male reproductive organs, two sets of shrimp samples were selected for analysis. Shrimp from different sources, namely wild-caught and domesticated shrimp, were used as representatives of shrimp with high and low total sperm counts, respectively. The effects of shrimp diets on total sperm counts were also examined as the use of polychaetes as live feed for male brooders has been shown to produce higher spermatophore weights and higher total sperm counts than the use of feed pellets [18, 20]. The results from this study are summarized in Fig 8. The comparative analysis of levels of eicosanoids and PUFAs in testes and vas deferens revealed that levels of 15d-PGJ$_2$, (±)8-HETE, and (±)12-HEPE in shrimp testes were lower than those in vas deferens in all shrimp samples (Fig 8A), suggesting that these eicosanoids may be essential for the sperm maturation process.

Eicosanoid and PUFA profiles were also compared for shrimp from different sources (wild-caught vs. domesticated shrimp; Fig 8B) and for shrimp fed with different diets (polychaete- and pellet-fed shrimp; Fig 8C). In both sets of samples, high levels of (±)12-HEPE, (±)18-HEPE, and EPA in testes as well as high levels of (±)12-HEPE in vas deferens were negatively correlated with total sperm counts (Fig 8B and 8C). In contrast, high levels of PGE$_2$ in vas deferens were positively correlated with high sperm counts in shrimp from both sets of samples. In humans, addition of PGE$_2$ and PGF$_{2\alpha}$ at low physiological levels to spermatozoa has been shown to improve sperm function [6]. Furthermore, transcriptomic analyses in crab gonads also provided supporting evidence regarding the positive effects of eicosanoid biosynthesis pathway in crustacean male reproductive maturation. This led to the identification of *prostaglandin E synthase 2* and *prostaglandin F synthase* as candidates for the regulators of growth, sexual differentiation, and reproduction in the testis of the orange mud crab *Scylla olivacea* [53]. Similarly, *prostaglandin E synthase* and *prostaglandin E2 receptor* were also identified as potential regulators of gonadal development in *P. trituberculatus* [54]. These data were also supported by a study in mammals, in which cyclooxygenase-2 and prostaglandin synthase enzymes that regulate the conversion of ARA to PGE$_2$ could serve as a local modulator of testicular activity in Leydig and Sertoli cells [55]. Therefore, we propose that eicosanoids also serve as modulators for testicular development and sperm maturation process in *P. monodon*. Our results not only expand the coverage of eicosanoid biosynthesis pathway in crustaceans, but also suggest that the roles of eicosanoids in spermatogenesis are conserved between crustaceans and mammals. Furthermore, the correlations between total sperm counts and high levels of eicosanoids in shrimp testis and vas deferens also suggest an alternative approach to improve total sperm counts by increasing the prostaglandin biosynthesis while suppressing the production of HEPEs in the male reproductive tract of penaeid shrimp.

## Supporting information

**S1 Table. Percentage of internal standards recovered from liquid-liquid extractions of *P. monodon* testes and vas deferens.**
(DOCX)

**S2 Table. Regression equations for the quantification of PUFAs and eicosanoids in *P. monodon*.**
(DOCX)

**S3 Table. Criteria for the identification of eicosanoids and PUFAs using retention time, precursor ion, proposed fragment ion, and m/z distribution.**
(DOCX)

**S1 File. Body length, body weight, spermatophore weight, and total sperm count of wild-caught and domesticated shrimp.**
(XLSX)

**S2 File. UHPLC-HRMS/MS analysis of testes and vas deferens of wild-caught and domesticated males.**
(XLSX)

**S3 File. Analysis of fatty acid profiles in polychaetes and feed pellets using GC-FID.**
(XLSX)

**S4 File. UHPLC-HRMS/MS analysis of testes and vas deferens of polychaete- and pellet-fed shrimp.**
(XLSX)

**S5 File. Sperm count and sperm abnormality in polychaete- and pellet-fed shrimp.**
(XLSX)

## Acknowledgments

We thank Ms. Somjai Wongtripop and Ms. Jutatip Khudet for their expertise in shrimp rearing. We thank Associate Professor Prapin Wilairat and Dr. Samaporn Teeravechyan for fruitful discussions.

## Author Contributions

**Conceptualization:** Suganya Yongkiettrakul, Rungnapa Leelatanawit, Wananit Wimuttisuk.

**Formal analysis:** Pisut Yotbuntueng, Surasak Jiemsup, Pacharawan Deenarn, Punsa Tobwor, Wananit Wimuttisuk.

**Funding acquisition:** Wananit Wimuttisuk.

**Investigation:** Pisut Yotbuntueng, Surasak Jiemsup, Pacharawan Deenarn, Punsa Tobwor, Kanchana Sittikankaew, Rungnapa Leelatanawit, Wananit Wimuttisuk.

**Methodology:** Surasak Jiemsup, Suganya Yongkiettrakul, Thapanee Pruksatrakul, Wananit Wimuttisuk.

**Project administration:** Wananit Wimuttisuk.

**Resources:** Vanicha Vichai, Nitsara Karoonuthaisiri, Wananit Wimuttisuk.

**Software:** Pisut Yotbuntueng.

**Supervision:** Suganya Yongkiettrakul, Vanicha Vichai, Wananit Wimuttisuk.

**Validation:** Pisut Yotbuntueng, Pacharawan Deenarn, Punsa Tobwor.

**Visualization:** Pisut Yotbuntueng, Pacharawan Deenarn, Punsa Tobwor.

**Writing – original draft:** Pisut Yotbuntueng, Suganya Yongkiettrakul, Wananit Wimuttisuk.

**Writing – review & editing:** Vanicha Vichai, Nitsara Karoonuthaisiri, Wananit Wimuttisuk.

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
