## [Decision Letter · Decision Letter 0]

18 Aug 2022

PONE-D-22-17666Differential distribution of eicosanoids and polyunsaturated fatty acids in the *Penaeus monodon* male reproductive tract and their effects on total sperm countsPLOS ONE

Dear Dr. Wimuttisuk,

Thank you for submitting your manuscript to PLOS ONE. After careful consideration, we feel that it has merit but does not fully meet PLOS ONE’s publication criteria as it currently stands. Therefore, we invite you to submit a revised version of the manuscript that addresses the points raised during the review process.

We look forward to receiving your revised manuscript.

Kind regards,

Gao-Feng Qiu

Academic Editor

PLOS ONE

Journal Requirements:

2. In your Methods section, please provide additional information regarding the permits you obtained for the work. Please ensure you have included the full name of the authority that approved the collection site access and, if no permits were required, a brief statement explaining why.

Reviewers' comments:

Reviewer's Responses to Questions

**Comments to the Author**

1. Is the manuscript technically sound, and do the data support the conclusions?

Reviewer #1: Yes

Reviewer #2: Yes

2. Has the statistical analysis been performed appropriately and rigorously? 

Reviewer #1: Yes

Reviewer #2: Yes

3. Have the authors made all data underlying the findings in their manuscript fully available?

Reviewer #1: Yes

Reviewer #2: Yes

4. Is the manuscript presented in an intelligible fashion and written in standard English?

Reviewer #1: Yes

Reviewer #2: Yes

5. Review Comments to the Author

Reviewer #1: I like this MS very much. By using ultra-high performance liquid chromatography coupled with Orbitrap high resolution mass spectrometry, the authors confirms the presence of PUFAs and eicosanoids in crustacean male reproductive organs, but also suggests that the eicosanoid biosynthesis pathway may serve as a potential target to improve sperm production in shrimp. This is a very original and valuable work in this domain. The technology used is correct and the MS text is clean and easy to understand.

Reviewer #2: In this study, the authors have evaluated the levels of eicosanoids and PUFAs in testes and vas deferens of shrimp Penaeus monodon and analyzed sperm counts between wild-caught and domesticated shrimp. This study could confirm the presence of eicosanoids in shrimp male and provide the helpful information for understanding the roles of eicosanoids in crustacean spermatogenesis.

1. The manuscript is well written. The introduction section is focused on giving us a good background of that topic.

2. The author should modify the sentence “High levels of dietary polyunsaturated fatty acids (PUFAs) showed a positive impact on crustacean sperm production (17,18)”. Reference 17 is focused on the effect of prostaglandin E2 on Penaeus monodon oocyte development in vitro (Menunpol et al., 2010), but not sperm production.

Reference: Meunpol O, Duangjai E, Yoonpun R, Piyatiratitivorakul S. Detection of prostaglandin E2 in polychaete Perinereis sp. and its effect on Penaeus monodon oocyte development in vitro. Fish Sci. 2010;76: 281–286. doi:10.1007/s12562-009-0208-8

3. Line 88-89, the authors suggest that the roles of eicosanoids in crustacean spermatogenesis are conserved relative to mammals. It would be interesting if the author could add more data to show how the eicosanoids affect spermatogenesis of shrimp.

4. Line 436 “Results from other studies”, references should be added.

6. PLOS authors have the option to publish the peer review history of their article (what does this mean?). If published, this will include your full peer review and any attached files.

Reviewer #1: No

Reviewer #2: No

---

## [Author Response · Author response to Decision Letter 0]

25 Aug 2022

Response: We have checked and made sure that the manuscript meets PLOS ONE’s requirements. 

2. In your Methods section, please provide additional information regarding the permits you obtained for the work. Please ensure you have included the full name of the authority that approved the collection site access and, if no permits were required, a brief statement explaining why.

Response: We have provided the permit information in the Materials and methods section “All experiments were approved by the Institutional Animal Care and Use Committee of the National Center for Genetic Engineering and Biotechnology, Thailand (Approval Code BT-Animal 13/2560)” (Line 92-94). This permit covered the shrimp rearing experiment, shrimp transport, shrimp dissection, and sample collection. No permit was required for the collection site access of wild-caught shrimp as they were purchased from local fishermen that would otherwise sell these broodstock to restaurants for human consumption. 

3. Please review your reference list to ensure that it is complete and correct. If you have cited papers that have been retracted, please include the rationale for doing so in the manuscript text or remove these references and replace them with relevant current references. Any changes to the reference list should be mentioned in the rebuttal letter that accompanies your revised manuscript. If you need to cite a retracted article, indicate the article’s retracted status in the References list and also include a citation and full reference for the retraction notice.

Response: We have checked the reference and made corrections based on the reviewer’s recommendations. 

Reviewers' comments:

Reviewer #1: I like this MS very much. By using ultra-high performance liquid chromatography coupled with Orbitrap high resolution mass spectrometry, the authors confirm the presence of PUFAs and eicosanoids in crustacean male reproductive organs, but also suggests that the eicosanoid biosynthesis pathway may serve as a potential target to improve sperm production in shrimp. This is a very original and valuable work in this domain. The technology used is correct and the MS text is clean and easy to understand.

Response: We thank the reviewer for the positive comments and words of encouragement. 

Reviewer #2: In this study, the authors have evaluated the levels of eicosanoids and PUFAs in testes and vas deferens of shrimp Penaeus monodon and analyzed sperm counts between wild-caught and domesticated shrimp. This study could confirm the presence of eicosanoids in shrimp male and provide the helpful information for understanding the roles of eicosanoids in crustacean spermatogenesis.

1. The manuscript is well written. The introduction section is focused on giving us a good background of that topic.

Response: We thank the reviewer for the positive comments. 

2. The author should modify the sentence “High levels of dietary polyunsaturated fatty acids (PUFAs) showed a positive impact on crustacean sperm production (17,18)”. Reference 17 is focused on the effect of prostaglandin E2 on Penaeus monodon oocyte development in vitro (Menunpol et al., 2010), but not sperm production.

Reference: Meunpol O, Duangjai E, Yoonpun R, Piyatiratitivorakul S. Detection of prostaglandin E2 in polychaete Perinereis sp. and its effect on Penaeus monodon oocyte development in vitro. Fish Sci. 2010;76: 281–286. doi:10.1007/s12562-009-0208-8

Response: We apologize for the oversight. The correct citation (Meunpol et al. 2005) was replaced in the revised manuscript. 

Reference: Oraporn Meunpol, Panadda Meejing, and Somkiat Piyatiratitivorakul. Maturation diet based on fatty acid content for male Penaeus monodon (Fabricius) broodstock. Aquaculture Research. 2005; 36, 1216-1225

3. Line 88-89, the authors suggest that the roles of eicosanoids in crustacean spermatogenesis are conserved relative to mammals. It would be interesting if the author could add more data to show how the eicosanoids affect spermatogenesis of shrimp.

Response: Thank you for your kind suggestion. We have reviewed the methods required to show the effects of eicosanoids on spermatogenesis, which involve cryosectioning, DNA staining, and counting differentiated germ cells in shrimp testes. However, we apologize that we cannot perform the suggested experiment as all of our testis samples were already frozen at -80�C. Additionally, it was not possible to start another feeding trial to collect new shrimp samples due to the unavailability of male broodstock from the shrimp rearing facility (SGIC) at this time.

As we were unable to test the effects of eicosanoids on spermatogenesis, we modified the sentence in line 88-89 to “…the roles of eicosanoids in regulating total sperm number in crustaceans are conserved relative to mammals” to better suit the data in this manuscript. Nevertheless, we will keep this suggestion in mind when designing experiments for our future studies. 

4. Line 436 “Results from other studies”, references should be added.

Response: We have added references number 18 and 20 at Line 437 at the end of the suggested sentence. 

Reference 18: Oraporn Meunpol, Panadda Meejing, and Somkiat Piyatiratitivorakul. Maturation diet based on fatty acid content for male Penaeus monodon (Fabricius) broodstock. Aquaculture Research. 2005; 36, 1216-1225

Reference 20: Rungnapa Leelatanawit, Umaporn Uawisetwathana, Jutatip Khudet, Amornpan Klanchui, Suwanchai Phomklad, Somjai Wongtripop, Pacharaporn Angthoung, Pikul Jiravanichpaisal, Nitsara Karoonuthaisiri, Effects of polychaetes (Perinereis nuntia) on sperm performance of the domesticated black tiger shrimp (Penaeus monodon), Aquaculture. 2014, 433: 266-275, https://doi.org/10.1016/j.aquaculture.2014.06.034.

---

## [Editor Report · Decision Letter 1]

12 Sep 2022

Differential distribution of eicosanoids and polyunsaturated fatty acids in the *Penaeus monodon* male reproductive tract and their effects on total sperm counts

PONE-D-22-17666R1

Dear Dr. Wimuttisuk,

We’re pleased to inform you that your manuscript has been judged scientifically suitable for publication and will be formally accepted for publication once it meets all outstanding technical requirements.

Kind regards,

Gao-Feng Qiu

Academic Editor

PLOS ONE
---

## [Editor Report · Acceptance letter]

14 Sep 2022

PONE-D-22-17666R1 

Differential distribution of eicosanoids and polyunsaturated fatty acids in the *Penaeus monodon* male reproductive tract and their effects on total sperm counts 

Dear Dr. Wimuttisuk:

I'm pleased to inform you that your manuscript has been deemed suitable for publication in PLOS ONE. Congratulations! Your manuscript is now with our production department. 

Kind regards, 

on behalf of

Prof. Gao-Feng Qiu 

Academic Editor

PLOS ONE